# Personal Safety and Fear of Sexual Harassment among Female Garment Workers in Bangladesh

**DOI:** 10.3390/ijerph182413406

**Published:** 2021-12-20

**Authors:** Muhammad Akram Uzzaman, Zamadonda Nokuthula Xulu-Kasaba, Muhammad Ehsanul Haque

**Affiliations:** 1Department of Psychology, Jagannath University, Dhaka 1100, Bangladesh; akrambro@gmail.com; 2Discipline of Optometry, School of Health Sciences, University of KwaZulu-Natal, Durban 4001, South Africa; XuluKasabaZ@ukzn.ac.za; 3Graduate School of Business, MANCOSA, Durban 4001, South Africa

**Keywords:** sexual crime, personal safety, female garment workers

## Abstract

Personal safety and fear of sexual harassment may discourage women from participating at work and in public life, limiting their life opportunities. The study proposed to determine personal safety and fear of sexual harassment among female garment workers in Bangladesh. This cross-sectional study was conducted among 201 female garment workers from Dhaka and Chittagong cities. Participants were selected using snowballing sampling techniques with the data collected by using anonymised questionnaires. The Pearson product–moment correlation and analysis of variance were employed using SPSS version 27.0. Results showed that 25% of the participants perceived that they were most likely to be sexually harassed by their manager and 25% never felt safe going to work. Age and the marital status of the participants were significantly associated with personal safety and fear of sexual harassment (*p* < 0.05). The correlation analysis found a significant positive correlation between personal safety and the fear of sexual harassment [r (201) = 0.85 **, *p* < 0.05], among the participants. A deep commitment from leadership with cooperation at all levels of the organisations is required to address these acts of violence and organisational conditions, rather than a form of unreflective compliance or a ‘gender-neutral’ approach that fails to recognise individual needs and maintain gender inequality.

## 1. Introduction

The World Health Organisation (WHO) defined sexual harassment as any unwelcome sexual behaviour or requests for sexual favours. These include verbal or physical conduct, or any gesture of a sexual nature, particularly behaviour construed as offensive or intimidating to the person concerned [1]. The International Labour Organisation (ILO) added two additional categories: ‘quid pro quo’ and ‘hostile working environment’. Quid pro quo sexual harassment is when a worker is threatened with the loss of employment when not acceding to the request for sexual favours. Harassment within a hostile working environment includes conduct in the workplace that would be viewed as intimidating or hostile [2]. A wide range of individuals such as teachers, colleagues, supervisors, subordinates, and third parties could perpetrate this type of activity [3]. Within educational settings, some male academics have considered it their right to demand sexual favours in return for grades or career progression that falls in line with the ILO definition of hierarchical and gendered power relations within this context or in occupational settings [4].

Bangladesh has its own rich culture and heritage. The people of Bangladesh have inherited a simple code of life. Irrespective of having such a great cultural heritage, over time it has developed some cultural backdrops such as showing the inferior attitude towards girls and women [5].

There is no unique definition of sexual harassment. For example, Kenny defines sexual harassment as a kind of dominance due to retaining power in terms of gender, and social position [6]. On the other hand, Gutek reported that sexual harassment is the unwanted and non-exchanging male attitude towards women in the form of sexual assertion [7]. Mahtab indicated that sexual harassment is a kind of “unwelcome sexual behaviour” that is offensive, humiliating, and dreadful. Sometimes it can be direct, indirect, verbal, and physical in nature [8].

The Prevention of Repression against Women and Children Act, 2000 in Bangladesh does not clearly articulate sexual harassment from a legal point of view. The act only includes the physical characteristics such as touching various bodily parts of a child or woman, making an impolite gesture, or annoying a woman sexually or verbally in the course of satisfying their sexual urge mischievously [9]. According to the High Court division of the Supreme Court of Bangladesh, sexual harassment is defined as some sexually suggestive comments or gestures conveyed directly or over the phone, sending an email in indecent means, watching indecently, and the use of some offensive slang languages such as “sundori”. Farhan in his newspaper report claims that because of lack of awareness, the educational institutes and the workplaces of Bangladesh have not coped with the guidelines of the court [10].

Personal safety is often misinterpreted or misunderstood and the lack of an unambiguous and widely accepted definition has resulted in the term being used in a variety of contexts. Waters, Neale, Hutson, and Mears conducted a Delphi method of research to define personal safety. Based on their findings, they determined that “Personal safety is defined as an individual’s ability to go about their everyday life free from the threat or fear of psychological, emotional or physical harm from others”. To enhance the understanding of personal safety, participants from the study added “Harm can be intentional or accidental and includes harm against your property and personal effects as well as against the person. Personal safety is freedom from perceived risk, actual risk, and fear, where risk is the likelihood of coming to harm. Personal safety is distinct from health and safety” [11].

Sexual harassment is a growing, global phenomenon with attributed costs [1]. Studies from high-income countries have shown that the effects of this form of workplace harassment lead to psychological distress such as anxiety, anger and depression, demotivation in job performance, and physical distress such as fatigue, weight loss, and even post-traumatic stress disorder [12,13,14].

Economic hardships due to job losses can occur when victims are compelled to relinquish their positions or are dismissed as penance for reporting crimes of this nature. Therefore, lost opportunities for career advancement are serious economic consequences of sexual harassment for the victim. For instance, organisations in which harassment is prevalent suffer from absenteeism, increased staff turnover, and lower job performance and productivity. These actions result in increased legal fees and a negative public image for the organisation [15].

Individuals who have previously been victims of crime are likely to feel unsafe as evidence from international studies has indicated, although some researchers have recommended that more in-depth studies are required [16,17]. Should a victim be a witness to an assault, particularly in the cases of family members, this would influence the individual’s level of personal safety.

Higher levels of insecurity are detected among the disadvantaged members of the community such as certain ethnic groups, the economically disadvantaged, elderly people, females, and vulnerable youth [17,18,19,20,21]. Although the youth are more likely to experience being victimised, the disabled and elderly tend to experience higher levels of fearfulness [18,19].

The importance of effective social contacts in the neighbourhood is of paramount importance to specific groups of women. According to Yates and Ceccato, this fear can lead to avoidance strategies and other behavioural changes [17]. Loukaitou-Sideris who investigated women’s fear of transportation environments reported that social disorganisation in the environment will generate fear when there is a lack of maintenance with abandoned buildings nearby with darkness, desolation, and no security monitoring [22].

The Bangladesh Labour Law has ensured safe and secure environments for women in their workplace [23]. However the shame and stigma associated with harassment, particularly sexual harassment, lack of exemplary punishment of rapists, and fear of reprisals at work prevent women from making formal complaints, despite the high prevalence. Most of the existing research has overlooked the fear of sexual harassment and the personal security of women workers from a psychological point of view. Considering these, the present study aimed to investigate the fear of sexual harassment and personal safety issues of female garment workers.

## 2. Materials and Methods

### 2.1. Research Design and Participants

This was a cross-sectional study conducted among 201 female garment workers. Participants were recruited using snowball sampling, due to the personal nature of this study, and were drawn from different slums within the Dhaka and Chittagong cities in the country of Bangladesh situated in Southern Asia, as most of the garment sewing factories/businesses are situated in these two cities.

### 2.2. Research Instrument

Fear of Sexual Crime Scale (FSCS) theory was developed by Gorman-Smith, Tolan, and for measuring the fear of being a victim of a violent crime in the home and/or neighbourhood, the impact of such fear, and steps taken to protect oneself from crime [24]. The scale contains 13 Likert-type items in which response options are: ‘not fearful’ = 1, ‘a little fearful’ = 2, ‘somewhat fearful’ = 3, ‘very fearful’ = 4, where the highest score indicates the greatest level of fear. The internal consistency of the scale is from 0.77 to 0.86. 

Personal Safety Scale (PSS) was developed by Henry for measuring feelings of safety in a neighbourhood or workplace and travelling between the two destinations [25]. The PSS has a total of eleven items resembling three points of Likert-type items. Point values are assigned as ‘never’ = 0, ‘sometimes’ = 1 and ‘always’ = 2. A higher score indicates a greater sense of safety by the respondent. The internal consistency of the scale was from 0.63 to 0.89. 

### 2.3. Data Collection

Data were collected using an anonymised self-administered questionnaire comprising three sections. The first section focused on socio-demographic information. The second posed 12 questions related to the fear of sexual harassment. The last section had 10 statements related to personal safety issues. A bilingual expert translated the research instrument for the participants initially from English to Bangla, and Bangla to English. The Bangla version of the questionnaire was pre-tested among 10 female garment workers from a different area to ensure optimal understanding of the instrument and to make changes if necessary. No amendments were made. The questionnaire was then piloted among 15 female garment workers from the slums within the study target area. The Cronbach’s alpha of the adapted scale was found to be 0.93 for FSCS and 0.92 for PSS in Bangladeshi culture which denotes that the adapted scale was highly reliable.

Participants were informed that the investigation was purely for academic purposes and assured that their responses to the questionnaires would be kept confidential. Thus, after giving their consent, the study instruments were administered to the selected participants.

### 2.4. Data Analysis

Data were coded, captured, and analysed using SPSS version 27.0. Independent sample *t*-test, one-way ANOVA, and correlation analysis were employed. *p*-values < 0.05 were considered statistically significant.

## 3. Results

A total of 201 women participated in the study. It was found that half of them (51%) were between the ages of 21 years and 30 years, and 51% had work experience of five years or less. More than two-thirds (71%) were married and 51% were working the night shift (Table 1). 

Regarding the personal safety of the female garment workers, it was found that a quarter of the women never felt safe going to work (25%), 30% never felt safe waiting outside the company they work for, a third of them never felt safe coming home during their lunch break, and 35% never felt safe on the road when returning home from work. Just over half of them (56%) mentioned that they always felt safe at home, over a quarter (27%) felt safe walking around the neighbourhoods, and 37% felt safe visiting their neighbours (Table 2).

A quarter of the female garment workers mentioned that they were frequently likely to be sexually harassed by their boss (25%), and 24% indicated this occurred on the way back from work. Twenty-eight per cent reported that they were often sexually harassed at night whilst going to work and returning home (Table 3).

Overall mean scores for personal safety and fear of sexual harassment were compared with regards to participants’ demographic variables. A participant’s age was significantly associated with their safety score and fear of sexual harassment. It was found that older participants had significantly higher mean scores compared to their younger counterparts (*p* < 0.05) (Table 4 and Table 5).

Overall mean scores for personal safety and fear of sexual harassment were compared with regards to participants’ work experience. It was found that participants with less than one year of experience had the lowest mean score for personal safety and sexual harassment (Table 6). ANOVA test found a significant mean difference in personal safety among the different working experience groups (*p* = 0.002). The years of experience was not found to be significantly associated with sexual harassment (*p* = 0.128) (Table 7).

It was found that day shift workers had a higher fear of sexual harassment and personal safety than the night shift workers. However, the differences were not found to be statistically significant (*p* = 0.26 and 0.29 respectively) (Table 8).

Table 9 shows that the mean score of fear of sexual crime among female garment workers in Dhaka city (M = 23.86; SD = 7.16) was lower than that of Chittagong city (M = 31.08; SD = 6.736). The results from the independent sample *t*-test indicated that the mean difference was significantly different (*p* = 0.01). Similarly, the average safety scores were significantly lower among the garment workers from Dhaka city (mean = 21.82) than that of Chittagong city (mean = 29.29) (*p* = 0.01).

The divorced women scored the highest mean score for fear of sexual crime (M = 36, SD = 2.83) and personal safety (M = 27.50, SD = 12.02) (Table 10). The ANOVA test shows that marital status (F = 8.85, *p* = 0.01) had a significant joint effect on the fears of sexual crime and personal safety of female garment workers (*p* = 0.01) (Table 11). 

Table 12 shows the Pearson product–moment correlation between personal safety and the fear of sexual crime. The results indicate that there was a significantly strong positive correlation existing between personal safety and fears of sexual crime [r (201) = 0.85 **, *p* < 0.05], that is, the more the insecurity the more the fear of sexual crime felt by garment female workers in their professional lives (Table 12).

## 4. Discussion

The present study’s purpose was to assess and compare the extent of fears regarding sexual harassment and the personal security of female garment workers between the Dhaka and Chittagong cities. Many female garment workers do not feel safe in the workplace or travelling there, or outside of the work environment. The findings are in line with another study conducted among female students in Bangladesh. The study found that most of the students were the victims of sexual harassment at least once in their lives and most of them were victimised outside of their educational institutions. The study also revealed that these incidents or experiences of their own and that of others have generated a considerable fear among them [26].

A global study revealed that students who displayed particular characteristics were more likely to be fearful of travelling than others. The same study reported gender as an overriding determinant of fear with females who expressed higher levels of fear and concern in transit environments than the male students [27].

The results depicted no significant difference between the day shift and night shift female garment workers in terms of their fear of sexual harassment and personal safety. These findings are in line with a previous study conducted among garment workers in Bangladesh [28]. However, another study conducted among 180 female garment workers in the Dhaka city of Bangladesh found that 33.3% of workplace violence occurred during day time, 37.8% during night time, and 28.9% during overtime [29]. Siddiqi investigated sexual harassment among industrial workers for strategies for intervention in the workplace and her study revealed that night shift women workers face tremendous sexual harassment compared to that of the day shift female workers. This resulted in embarrassment, inability to concentrate on work, declines in productivity, and other mental health problems (e.g., anxiety, fear and depression) [30].

Age is known to be a significant predictor of sexual harassment. The present study also found this aspect to be significantly associated with sexual harassment. Another study conducted among university students found that age alone can be a strong predictor of sexual victimisation among female students [31,32]. It is also possible that individual characteristics interact and influence women’s vulnerability to crime and fear of crime [33].

Another finding in this study indicated significant differences between the female garment workers of Dhaka and those in the Chittagong city in terms of their fear of sexual harassment and personal safety. That means that the female garment workers of Dhaka City face comparatively low risks of sexual harassment and safety problems than that of the workers of Chittagong city. The security infrastructure in Dhaka city is much stronger and more effective than in Chittagong city, which lacks a sound safety system [34]. Due to overcrowding in Dhaka, sexual harassment instigators sometimes do not obtain opportunities to commit such acts in public spaces. Human rights organisations are also very active in Dhaka. Due to the launch of separate bus services for women in Dhaka, women do not face any difficulties while on the bus. Nevertheless, female employees within the garment sector in Dhaka sometimes face different types of harassment, which concurs with previous studies. For example, Sohani et al. conducted a study that investigated the patterns of workplace violence against female garment workers in a selected area of Dhaka city and reported that 68.9% of the workers there experienced psychological harassment with 7.2% reporting that they were victims of sexual harassment [28].

The present study found a significant association between marital status and fears of sexual harassment and personal safety. This means that divorced and separated female workers faced more sexual harassment and security problems in their everyday lives. These groups of women may stay away from their husbands or family due to quarrelling, poverty, distant working environments, maladjustment such as divorce or separation, etc. Perpetrators (such as business owners, managers, peer workers, local miscreants, gatemen, and drivers) take advantage and try to establish sexual relations with female garment workers. This finding is consistent with the findings of a study conducted by Action Aid [35]. The study revealed that if a perpetrator failed to execute sexual harassment, then they threatened the subordinate with a transfer, a reduced salary, or even humiliation in the workplace [35].

Workplaces have been identified as social spaces where change that addresses and seeks to eliminate sexual violence perpetrated against women can be implemented and enforced through gender equity measures [36]. Yet this form of violence remains rife and poorly addressed, despite legal interventions that have made sexual violence a criminal offense, and the widespread implementation of diversity management and training [37]. More than a third of the female garment workers in this study mentioned that they are frequently, or very often likely to be sexually harassed by their line manager. An Australian study that investigated sexual harassment in Australian workplaces revealed that 33% of the employees experienced sexual harassment at work in the last five years [38]. The same study found that 25% of the perpetrators of sexual harassment in the workplace were the victim’s or survivor’s direct manager or supervisor [38].

Correlational analysis showed a significant positive relationship between sexual harassment and the fear of security. It is possible that the women who face sexual harassment create and develop phobias, and experience security problems, anxiety, depression, emotional trauma, loss of employment, reputational issues, late payments on their bills, and fewer promotional opportunities. They might also develop inferiority complexes as a result of these problems [39]. Further to this, they do not venture outside comfortably, their movements are restricted and they suffer cognitive dissonance and maladjustment in their personal lives. These women also struggle with mental stress emanating from personal insecurity. This finding is similar to the findings of other studies conducted among garment workers in Bangladesh as well as workplace harassment in other Asian countries such as India, Nepal, and Sri Lanka [40,41].

### Implications

The findings of the present study could contribute to social science-oriented disciplinary and interdisciplinary epistemology and the corresponding policy practices of sexual harassment concerning female garment workers in Bangladesh. This study would be particularly useful in its role to create awareness amongst male and female garment workers through NGO representatives, human rights activists, policymakers, developmental psychologists, and the like who could empower them on the risks and challenges of sexual harassment and personal insecurity, and how to resolve such issues. If the concerned authorities follow through and develop strategies to mitigate issues, then the rate of sexual harassment and personal insecurity should gradually reduce in the future within the garment sector. As a result, the women workers’ lives will be safer and they would be able to contribute meaningfully to their families, personal life, and ultimately to society for national development.

The present study is not without limitations. The sample size is relatively small when compared with the general population, so it would be difficult to generalise these findings. All the respondents were selected only from the Dhaka and Chittagong cities and were not representative of all the areas of Bangladesh. Therefore, this finding is not fully representative of Bangladesh, and the results should be interpreted with caution. Thirdly, some respondents might not have answered the questions truthfully due to the sensitive nature of the topic.

## 5. Conclusions

Results show that levels of sexual harassment are associated with levels of perceived safety in these settings. The study provided indications and a better understanding concerning the fears that female garment workers might feel daily.

To improve personal safety and reduce sexual harassment in the workplace it should become common practice to prioritise legislation across organisations championed by strong leadership. Without a clear organisational approach to personal safety and sexual harassment, each worker’s response to a threatening or confrontational situation will vary significantly and be heavily dependent on individual judgement. To achieve consistent, effective personal safety in the workplace and reductions in sexual harassment, each worker must know how to identify, assess, reduce, and manage, the risk of violence and aggression.

## Figures and Tables

**Table 1 ijerph-18-13406-t001:** Summary of demographic information [*n* = 201].

Variables		Frequency	Percent
Age group	≤20 years	56	27.9
	21–30 years	102	50.7
	31–40 years	43	21.4
Years of experience	<1 year	43	21.4
	1–5 years	103	51.2
	>5 years	55	27.4
Job place	Dhaka	100	49.8
	Chitagong	101	50.2
Marital status	Married	143	71.1
	Unmarried	52	25.9
	Divorced	2	1
	Separation	4	2
Office time	Day	99	49.3
	Night	102	50.7
	Total	201	100

**Table 2 ijerph-18-13406-t002:** Distribution of responses regarding personal safety [*n* = 201].

Statement	Never	Sometimes	Always
I feel safe going to work.	24.9	33.8	41.3
I feel safe when waiting outside the company.	29.9	24.9	45.3
I feel safe while at the workplace.	16.4	30.8	52.7
I feel safe during lunch break when I come home.	32.8	36.8	30.3
I feel safe everywhere in the workplace.	25	27.5	47.5
I feel safe on the road when coming back home from work.	35	40.5	24.5
I feel safe in my own home.	8	35.8	56.2
I feel safe walking around my neighbourhood.	30.8	41.8	27.4
I feel safe visiting my neighbours.	21.4	41.8	36.8
I feel safe walking on the road or park next to my workplace.	33.3	44.3	22.4
I feel safe walking with my colleagues.	13.5	46	40.5
I feel safe while shopping in my neighbourhood.	23.4	48.3	28.4

**Table 3 ijerph-18-13406-t003:** Summary of statements related to sexual harassment [*n* = 201].

Statement	Very Often	Often	Sometimes	Never
How likely are you to be sexually harassed by your colleagues?	20.4	10	22.9	46.8
Overall, how likely are you to be sexually harassed at the workplace?	13.4	18.4	25.4	42.8
How likely are you to be sexually harassed by your neighbours?	8	18.9	32.3	40.8
How likely are you to be sexually harassed at home by someone?	4.5	10.9	33.3	51.2
How likely are you to be sexually harassed on the way to work?	11.9	33.3	37.8	16.9
How likely are you to be sexually harassed on the way back from work?	23.9	21.4	36.3	18.4
How likely are you to be sexually harassed by your boss at the workplace?	25.4	10.9	23.9	39.8
How likely are you to be sexually harassed on the way to and back from work during the day?	4.5	31.8	48.8	14.9
How likely are you to be sexually harassed on the way to and back from work at night?	27.9	47.8	17.9	6.5
Overall, how likely are you to feel like a victim of sexual harassment?	18.4	23.4	44.3	13.9

**Table 4 ijerph-18-13406-t004:** Mean distribution of personal safety and sexual harassment regarding the age of the participants [*n* = 201].

Age Group		*n*	Mean	Std. Deviation
PSS	≤20 years	56	21.88	7.611
	21–30 years	102	26.70	6.085
	31–40 years	43	27.72	5.030
	Total	201	25.57	6.738
FSH	≤20 years	56	23.89	8.418
	21–30 years	102	28.56	7.785
	31–40 years	43	29.63	5.278
	Total	201	27.49	7.819

**Table 5 ijerph-18-13406-t005:** ANOVA test output for personal safety and sexual harassment with regards to the age of the participants.

		Sum of Squares	df	Mean Square	F	*p*-Value
PSS	Between Groups	1092.849	2	546.425	13.544	<0.05
	Within Groups	7988.355	198	40.345		
	Total	9081.204	200			
FSH	Between Groups	1037.668	2	518.834	9.182	<0.05
	Within Groups	11,188.551	198	56.508		
	Total	12,226.219	200			

**Table 6 ijerph-18-13406-t006:** Mean distribution of personal safety and sexual harassment regarding work experience.

Age Group		*n*	Mean	Std. Deviation
PSS	<1 year	43	22.40	6.839
	1–5 years	103	26.32	6.537
	>5 years	55	26.65	6.386
	Total	201	25.57	6.738
FSH	<1 year	43	25.40	8.452
	1–5 years	103	27.85	7.458
	>5 years	55	28.44	7.819
	Total	201	27.49	7.819

**Table 7 ijerph-18-13406-t007:** ANOVA test output for personal safety and sexual harassment with regards to work experience.

		Sum of Squares	df	Mean Square	F	*p*-Value
PSS	Between Groups	556.061	2	278.031	6.457	0.002
	Within Groups	8525.143	198	43.056		
	Total	9081.204	200			
FSH	Between Groups	251.597	2	125.799	2.080	0.128
	Within Groups	11,974.622	198	60.478		
	Total	12,226.219	200			

**Table 8 ijerph-18-13406-t008:** Independent sample *t*-test of fear of sexual crime and personal safety as per shift work.

Office Time		*n*	M	SD	t	df	*p*-Value
FSH	Day	99	28.12	8.33	1.13	199	0.26
	Night (Shift)	102	26.87	7.27			
	Total	201					
PSS	Day	99	26.08	6.74	1.05	199	0.29
	Night (Shift)	102	25.08	6.72			
	Total	201					

**Table 9 ijerph-18-13406-t009:** Independent sample *t*-test of fear of sexual crime and personal safety based on geographical location.

Geographical Location		*n*	M	SD	t	df	*p*-Value
FSS	Dhaka city	100	23.86	7.16	7.36	199	0.01
	Chittagong city	101	31.08	6.74			
	Total	201					
PSS	Dhaka City	100	21.82	6.37	9.42	199	0.01
	Chittagong City	101	29.29	4.74			
	Total	201					

**Table 10 ijerph-18-13406-t010:** Mean and standard deviation of fear of sexual crime and personal safety according to the marital status of the participants.

Marital Status	Fear of Sexual Crime			Personal Safety		
	*n*	M	SD	*n*	M	SD
Married	143	28.92	7.41	143	26.98	6.08
Unmarried	52	23.10	7.22	52	21.50	6.59
Divorced	2	36	2.83	2	27.50	12.02
Separation	2	29	10.03	2	27.25	10.06

**Table 11 ijerph-18-13406-t011:** ANOVA output for fear of sexual crime and personal safety by marital status.

	Source	SS	df	MS	F	Sig
Sexual harassment	Between Group	1451.55	3	483.85	8.85	0.01
	Within Group	10,774.68	197	54.69		
	Total	12,226.22	200			
Personal safety	Between Group	1164.02	3	388.01	9.66	0.01
	Within Group	7917.19	197	40.19		
	Total	9081.20	200			

**Table 12 ijerph-18-13406-t012:** Correlation between personal safety and fear of sexual crime.

Variable	Fear of Sexual Crime
Personal Safety	0.85 **

** *p* < 0.05.

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
