# Peer review of "Personal Safety and Fear of Sexual Harassment among Female Garment Workers in Bangladesh"

_ijerph, 2021, doi:10.3390/ijerph182413406_

Round 1

Reviewer 1 Report

General comment.  Putting a number rather than the name of an author being quoted is not useful.  Please put name and date in this circumstance.

Line

34 – you say two extra categories – but only discuss one.  What is the other one?

44-45 slightly confused sentence – instead of typically leads to put includes – makes it flow

50 – 51 – one of the serious economic consequences?  And (51) why have oyu included perpetrator – you don’t explain how they have economic consequences.  I think you mean employer.

56 reference your ‘international researchers’

57 put 9-10 in the same brackets

65-69.  This paragraph lacks context and is too summarised to make sense.  It needs to be expanded to sense in the context of your article.  End of 69 an example of the need for a names researcher.

75-78 This sentence doesn’t quite make sense yet

82 – Cities not city

84/90 – use. Name not a number please

  1. many garment workers – not all garment workers

215 in the workplace

222 – more self aware etc then when – in comparison to what?

225 – women in the garment industry are women in Bangladesh.  This needs rewording – also – how do you know women in Bangladesh are more resilient now?

226 – 231 – I think this info – this research – should go in your introduction – and then in this section compare your results to theirs/. Also – use names not numbers for the researchers.  Ditto age…

  1. What hypothesis?

265 – 268 – you haven’t made any link that I can see with this finding

276 – 30% of whom?  Women? Garment workers (Australia)

293 what is a plan executor?

300 – nowhere in all of this do you talk about addressing men’s perpetration of abuse – why is it women who have to learn new skills etc to mitigate harm?

The conclusions section is a bit muddled and requires rewriting for clarity – also this is the only time you discuss male responsibility/change

Author Response

Dear Reviewer,

Thank you so much for your review on our manuscript. It is much appreciated.. we have revised the manuscript according to your valuable comments. Please see the attachment.

Thank you.

Reviewer 2 Report

This is in important topic and your findings are important but you need to do more work around defining sexual harassment - we are usually talking about things that happen at work not things that happen at home or that the neighbours do. You also need to define  what you mean by 'personal safety',  'sexual crime', sexual victimisation' - you seem to be all over the place with these and conflating a few ideas which is confusing to the reader.

Your title might be rearranged to be 'Personal Safety and fear of sexual harassment among female garment workers in Bangladesh' - the other way reads like workers are also fearful of personal safety. In your abstract you would speak about being fearful for personal safety, not of it. also, is 'choice' the right word here - do you really mean decision, ability, or opportunity or all of these?

There are a lot of references in the text where you stop the sentence without elaborating on what the reference says or means, e.g..'[9] and [10],  [15], [17] - you need to provide more information.

In your results you also need to talk about the higher percentages of women who never experience sexual harassment (see Table 3) - you seem to have ignored these quite positive findings - e.g. that 46.8  claim never sexually harassed by colleagues. In other words, a better and more thorough analysis of your article would significantly improve it.

In your discussion section you claim 'female garment workers do not feel safe to go to work [do you mean travelling to work?], the workplace [same thing as 'going to work' unless you mean 'travelling'] and outside of the workplace.' I don't think the results actually bear this out, so a little more attention to results, the details, etc., would be helpful.

This is an important paper so you need to turn your mind to further developing the ideas and content to improve its value to the reader and to the literature.

Author Response

Dear Reviewer,

Thank you so much for reviewing our manuscript. We have revised the manuscript according to your valuable comments. Please see the attachment.

Best regards

Prof. Hoque

Reviewer 3 Report

The authors are advised to provide further information concerning the topic’s cultural, moral, and legal facades; some references to Bangladesh authors would also be welcome. Additionally, you could describe your objective in more detail and provide some research questions. I do not know whether the Int. J. Environ. Res. Public Health allows one to simply refer to numbers instead of using the authors’ names, therefore, please insert the names of each reference as well, e.g., the report by Makizako et al. [34].   instead of «required [9] and [10] … according to [15] …» etc.

Materials and Methods

The authors are requested to completely reorganize this section by adding subtitles (Sample design or Participants, Procedure, Measures, and Statistics Analysis), as well as the relevant description in each paragraph.

Results

Please insert n and % in each table, where simple distributions are presented.

I wonder if the authors ought to present their results in one or two models of linear regression analysis, instead of using this plethora of tables with one-way, correlations etc. 

Author Response

Dear Reviewer,

Thank you so much for reviewing our manuscript. We have revised the manuscript based on your valuable comments.

Kind regards

Prof. Hoque

Round 2

Reviewer 2 Report

Thank you for the significant improvements to your paper.

I am still concerned that your paper may confuse readers for whom 'sexual harassment' is necessarily associated only with work, employment, or the provision of services etc. (see for example, the case in Australia, https://humanrights.gov.au/quick-guide/12096 and https://www.un.org/womenwatch/osagi/pdf/whatissh.pdf) , see also  https://www.legislation.gov.au/Details/C2014C00002   including at s28A). For many readers, sexual harassment is not about other harmful or offensive conduct of a sexual nature that may happen at home, be perpetrated by neighbours, or on the way to or from work (or anywhere else, such as on the way to the shops or to a friend's home, etc) but which may not cross the line sufficiently to become a criminal act or a crime (e.g. sexual assault or rape).  I think your article would be much improved if these variations in understanding and definitions were addressed (particularly if they are grounded in cultural differences) - similar behaviour at home might be understood in other places as being intimate partner violence - which depending on the actual act may be may not be criminal or unlawful - or may be, for example, rape or sexual assault in marriage or by a neighbour, or on the street. Cat-calling may not be a crime but it is sexual and offensive, but probably not sexual harassment unless there is some connection with work, employment, or the job. For many readers, it is the connection with work and the power imbalance between workers and employers, which may be the most significant. I think your paper would be better for identifying and addressing these differences.

I also found some sources which may be relevant to your work and could perhaps be referenced as some seems very on point.

https://actionaid.org/sites/default/files/publications/ActionAid%20briefing%20paper%20on%20Bangladesh%20garment%20workers%20FINAL.pdf

https://actionaid.org/publications/2019/sexual-harassment-and-violence-against-garment-workers-bangladesh

Sexual Harassment and Professional Women in Bangladesh

  • September 2003
  • Asia Pacific Journal on Human Rights and the Law 4(2):52-69
DOI:10.1163/1571815032120004 Authors: Shahnaz Huda
  • University of Dhaka

https://www.dhakatribune.com/bangladesh/2020/08/29/survey-most-women-face-sexual-harassment-at-work

Good luck.  

Author Response

Dear Reviewer,

Thank you so much for your valuable input. We hope that we have addressed your comments adequately.

Thank you.

Prof. Hoque

Reviewer 3 Report

none

Author Response

No suggestions were provided by the reviewer.